# Academic Perfectionism, Psychological Well-Being, and Suicidal Ideation in College Students

**DOI:** 10.3390/ijerph20010085

**Published:** 2022-12-21

**Authors:** Olga Fernández-García, María Dolores Gil-Llario, Jesús Castro-Calvo, Vicente Morell-Mengual, Rafael Ballester-Arnal, Verónica Estruch-García

**Affiliations:** 1Department of Developmental and Educational Psychology, Faculty of Psychology, University of Valencia, 46010 Valencia, Spain; 2Department of Personality, Assessment, and Psychological Treatments, Faculty of Psychology, University of Valencia, 46010 Valencia, Spain; 3Department of Basic and Clinical Psychology and Psychobiology, Faculty of Health Sciences, Jaume I University, 12007 Castellón de la Plana, Spain

**Keywords:** academic perfectionism, psychological well-being, suicidal ideation, college students, academic performance, gender differences, mental health

## Abstract

High levels of perfectionism in college students can compromise their academic performance and psychological well-being. This study aims to analyze the implication of perfectionism in psychological well-being and suicidal ideation in the last year. A total of 1.287 students from different degrees reported their academic performance in the previous academic year and completed questionnaires on academic perfectionism, psychological well-being, and suicidal thoughts in the last year. In both men and women, academic perfectionism correlates positively with academic performance and negatively with the different dimensions of psychological well-being. Likewise, both the average grade in the previous year (*β* = 0.364) and the level of demand of the degree (*β* = −0.461) are mediating variables between perfectionism and psychological well-being. Furthermore, the interaction between perfectionism and academic performance is related positively to psychological well-being, but not to with suicidal ideation. So, the relation between suicidal ideation and perfectionism is positive (OR = 1.075), but this is negative with academic performance (OR = 0.900), although both variables show a mild predictive capacity. These findings suggest that the levels of perfectionism are associated differently with the mental health of students, since if perfectionism is effective (high academic performance), psychological well-being is high, although in our results, this interaction is not as important for suicidal thoughts.

## 1. Introduction

Perfectionism is generally considered to be a desirable quality, particularly in the academic field, as it is often related to a higher performance, which is linked to personal and social success [1]. Perfectionism as a personality trait has been associated with motivational, cognitive, affective, and behavioral aspects which predict a good academic performance [2]. Meticulousness, persistence, and the need to demonstrate superiority are characteristics of more perfectionistic individuals that direct their behavior towards a better performance and higher academic goals. In this way, higher perfectionism scores are associated with a better self-efficacy and higher studying rates [3], all of which benefit performance. A separate discussion is whether these personality variables (meticulousness, persistence, and the need to demonstrate superiority) may compromise success when exceeding certain levels of concern and self-demand. In this regard, some studies suggest that people with high perfectionistic worries experience feelings of helplessness and a lack of control that could lead to a lower academic performance [4].

The other explanatory factors for a good performance are environmental factors. In this sense, it has been found that the students’ learning environment partly explains perfectionist behaviors [5]. Bußenius and Harendza [6] reported that students at the Hamburg Medical School showed very high levels of perfectionism and that this variable was found to be maladaptive and correlated with feelings of distress, worsening their performance. Therefore, it appears that high levels of performance are associated with perfectionism through personality and contextual variables, provided that the degree of perfectionism is not extremely high, as their performance is then compromised. Consequently, it may not be a desirable quality, especially if it is beyond the individual’s control.

According to a recent meta-analysis [7], the levels of perfectionism among young people, in general, have steadily increased over the past 27 years. Perfectionism is a multidimensional, intra-, and interpersonal construct consisting of extremely high personal standards or performance expectations and overly critical evaluations of oneself [2]. Perfectionistic individuals adopt a cognitive style based around the absolutist dichotomous all-or-nothing mindset and a willingness to avoid error [8], which is associated with an increased predisposition to psychological problems and poor positive psychological functioning. The research based on the study of perfectionism as a maladaptive condition has reported that more perfectionistic students present higher levels of fatigue, anxiety, depression, and hostility.

Based on the established evidence, feelings of worthlessness and harsh self-criticism associated with failing to live up to one’s expectations lead to a poorer adjustment and negative emotional states, due to the cognitive and emotional self-regulation strategies that trigger perfectionistic thoughts. More specifically, Hewitt and Flett [9] proposed that perfectionistic behavior can generate stress that results, partly, from the tendency to perform rigorous evaluations, focus on the negative aspects of your performance, and experience a low satisfaction. This idea, which has been subsequently supported [10], suggests that perfectionism can generate stress that, in turn, is associated with poor psychological well-being in such a way that pressure would act as a moderating variable between perfectionism and emotional well-being.

Psychological well-being is defined as the perception and evaluation that individuals have of their lives and, according to Ryff [11], it is a multidimensional concept determined by a subjective experience and is constituted by six interrelated dimensions. Self-acceptance implies the positive evaluation of oneself, positive relations with others involve the development of quality relationships, autonomy is associated with the feeling of self-determination, environmental mastery refers to the ability to manage your life and the world around you, purpose in life consists of discovering the meaning of life, and personal growth involves the feeling of continuous development. Greater subjective well-being would be linked to high levels of positive emotions, such as life satisfaction, happiness and pleasure, and low levels of negative emotions, such as depression and anxiety. According to a recent study investigating the relationship between perfectionism, academic resilience, and subjective emotional well-being in Korean college students [5], subjective well-being was negatively related to perfectionism.

Furthermore, some research conducted with gifted students [12] has suggested that perfectionism leads to severe self-criticism, self-doubt, and a fear of failure, which implies a greater tendency to develop suicidal thoughts and behaviors [13]. Suicide is a major public health problem with significant consequences, as almost 1 million people die from suicide every year worldwide [14]. Several theories have been developed to explain this positive correlation between perfectionism and suicidal behaviors. On the one hand, the Perfectionism Social Disconnection Model [15] states that perfectionism not only has a direct effect on suicidal tendencies, but it also has an indirect effect by creating or increasing the feeling of detachment from the social world. On the other hand, the Interpersonal Theory of Suicide (IPTS) states that suicidal ideation is associated with the perception of a burden and feelings of frustrated belongingness in meaningful interpersonal relationships more present in perfectionistic people. As determined by these theories, although suicide is rarely attributed to a single cause, perfectionistic traits appear to be a vulnerability factor. Research by the Alaska Injury Prevention Center et al. [16], in which the relatives of people who committed suicide were interviewed, provide evidence of the perniciousness of perfectionism, as 56% and 68.1% of the relatives, respectively, defined the deceased as perfectionists. Despite the above, the limitations and inconsistencies found in the non-empirical research published on this subject do not allow for the confirmation of the existence of a consistent relationship between the dimensions of perfectionism, suicidal ideation, and suicide attempts.

Regarding the influence of certain sociodemographic variables, the literature on perfectionism has paid very little attention to the existence of gender differences in this dimension. However, the few studies that address this point out that gender modulates the expression of specific dimensions of perfectionism (such as perfectionism as a trait or the self-representation of perfectionism) [17], with males generally presenting higher standards of excellence and scoring higher on perfectionism [18]. There does not seem to be as much consensus on the impact that this variable may have on the relationship between academic perfectionism and mental health: while some studies conclude that gender does not modulate this relationship, others argue that gender could enhance it.

Despite all of the above, recent studies suggest that perfectionism can be adaptive and be associated with a more positive emotional state, lower anxiety, higher self-esteem and, in some cases, better academic integration. Thus, positive perfectionism would refer to the set of cognitions and behaviors that direct individuals to achieve high-level goals through positive reinforcement and motivation to succeed. In contrast, negative perfectionism would represent individuals who struggle to attain unrealistic performance standards and includes negative reinforcement and a fear of failure. Therefore, positive perfectionism is considered to be a healthy and desirable characteristic, positively affecting psychological well-being, decreasing psychological distress, and making individuals feel joy and satisfaction with their life [19]. This new conceptualization that attributes a positive functionality of perfectionism should be considered.

### The Present Study

Being aware of the risks associated with fragile mental health in university students and the relevant role that perfectionist behaviors play in academic performance and psychological well-being, it was proposed to study these aspects in a sample of university students of degrees with different levels of demand. This study has multiple aims:

(a) The first one is to analyze the relationship between perfectionism, academic performance, emotional well-being, and suicidal ideation in the last year. Given the results obtained so far, it is expected that perfectionism correlates positively with performance and negatively with mental health dimensions.

(b) The second aim is to analyze whether two aspects moderate the relationship between perfectionism and mental health: one of a contextual nature (specifically, the general level of demand of the academic degree being studied) and another of personal perception (the level of performance achieved). Thus, the starting hypothesis is that the impact of high perfectionism on the reduction in mental health will be higher when studying more demanding degrees (e.g., medicine) and when academic performance is lower (i.e., when students perceive that their performance is lower than that of their degree).

(c) A final aim for this study would be to analyze whether gender plays a role in the relationship between academic perfectionism and mental health. In this sense, taking into account the discrepancies in the existing studies regarding the role of gender in the relationship between perfectionism and mental health, it is difficult to set a hypothesis; however, this last aim may help to clarify this topic of great interest.

## 2. Materials and Methods

### 2.1. Procedure and Participants

Participants were recruited online during the 2019–2020 academic year. The contact was established via email and social networks with groups of students who were studying bachelor’s, master’s, or doctoral degrees at universities in the province of Valencia (eastern Spain), providing them with information about the study and requesting their collaboration in the dissemination of the questionnaires. More specifically, contact was made via email with males and females studying medicine (1278 emails sent), psychology (1005), education (937), nursing (1144), biochemistry and biomedical sciences (321), physics (681), and odontology (836) at the University of Valencia, whose email addresses had been obtained through the university’s public search server (which allows for filtering by degree and role). Moreover, using the research group’s social networks, contact was made with students too. As a result, the data collected through social networks was approximately 10%, while 90% of the participants received the survey by email. In the specific case of contacts established by email, a response rate of approximately 22% was achieved.

As for the evaluation, the data were collected through a set of instruments administered through the LimeSurvey online platform, which allowed us to improve our compliance with ethical principles during the evaluation (informed consent is required to access the survey and the IP address of the device from which the user accesses the survey is not recorded). From the beginning, the participants were informed of the voluntary nature of their participation in the study and the anonymity and confidentiality of the data provided. The participants did not receive any remuneration.

Through this procedure, a total of 1415 participants were recruited. Of these, those who were not studying university studies (*n* = 20) or who had not completed at least 60% of the questionnaires (*n* = 108) were excluded. After this screening, the final sample for the study consisted of a total of 1287 university students. The mean time spent completing the survey was 9.06 min (*SD* = 3.86).

The study complies with the ethical principles of the Declaration of Helsinki and was approved by the Ethics Committee of the University of Valencia. The authors have no competing interests to declare and certify responsibility.

### 2.2. Measures

#### 2.2.1. Socio-Demographic Information

The participants completed an ad hoc instrument where the following information was collected: (1) sex (*female/male/non-binary*), (2) age (*18–21 years old/22–25 years old/26–29 years old/30–33 years old/over 33 years old*), and (3) whether they had completed their studies or were currently studying (open response).

#### 2.2.2. Academic Performance

The academic performance of the participants was measured through the average grade obtained during the last academic year. The participants were asked: “*What was your average grade in the last academic year?*”. For the response options, the participants had to indicate within which range their average grade was (ten response options [between 5 and 5.4 until between 9.5 and 10]).

#### 2.2.3. Level of Demand of the Degree

Given that there is no direct indicator of the level of demand for a degree, in this study the “admission cut-off score” was taken as an approximate measure of this construct. In Spain, due to the limited number of university places, students access their desired degrees according to a score that includes both the average performance during their pre-university education and the grade obtained in an official university entrance exam held at the state level. The range of this score is between 0 and 14. The most demanding degrees are only accessible to students with the highest grades.

#### 2.2.4. Academic Perfectionism

The *Study-related Perfectionism Scale* (SPS) [3] is a self-report instrument composed of 11 items that allows for the assessment of academic perfectionist attitudes towards studying. It is answered using a Likert-type scale with five response options ranging from 1 (“Strongly disagree”) to 5 (“Strongly agree”). Regarding its psychometric properties, this instrument has shown a very good internal consistency (α = 0.80).

#### 2.2.5. Psychological Well-Being

The *Brief Scale of Psychological Well-Being for Adolescents* (BSPWB-A) [20] consists of 20 items that are answered on a Likert-type scale with six response options ranging from 1 (“Strongly disagree”) to 6 (“Strongly agree”). It allows for the assessment of 4 of the 6 dimensions of psychological emotional well-being established by Ryff: (a) self-acceptance (positive evaluation of oneself and one’s past life and acknowledgement and acceptance of good and bad qualities), (b) positive relations with others (having quality relationships, concern for the well-being of others, and understanding of the give and take of human relationships), (c) autonomy (sense of self-determination and ability to resist social pressures to think and act in certain ways), and (d) personal growth (sense of continuous development as a person and openness to new experiences for self-knowledge and self-improvement). In the present study, the reliability of both the total scale (α = 0.89) and the subscales (α between 0.76 and 0.86) was adequate.

#### 2.2.6. Suicidal Ideation

The suicidal ideation was measured through a single question (“*Have you ever been so distressed that you have thought of death as liberation?*”) of dichotomous response (Yes/ No). At the beginning of the survey, the participants were informed that the questions were referred to last year.

### 2.3. Stadistical Analysis

Data were analyzed using the SPSS Version 25.0 statistical package and the G*Power software version 3.1 (calculation of the effect size). First, descriptive analyses were performed to classify the participants in relation to their sociodemographic characteristics and the variables under study (perfectionism, performance, level of demand of the degree, and psychological well-being). Likewise, the possible existence of significant differences according to sex was analyzed. These differences were evaluated using t-tests (continuous variables) and Chi-square comparisons (categorical variables). The effect size in these comparisons was estimated from Cohen’s d and Cramer’s V. For Cohen’s d, the effect sizes of around 0.20 were considered small, around 0.50 moderate, and above 0.80 large [21]; for Cramer’s V, these sizes corresponded to values of 0.10, 0.30, and 0.50, respectively. Subsequently, the degree of relationship between the main variables included in the study was estimated through a correlation analysis (Pearson’s r). Finally, a hierarchical linear regression and a logistic regression were performed through the intro method; the objective was to analyze the predictive capacity of perfectionism (independent variable [IV]) on psychological well-being (dependent variable [DV] in the linear regression) and on suicidal ideation (DV in the logistic regression). In order to analyze their possible moderating influence on the relationship between academic perfectionism and well-being/suicidal ideation, the following interaction terms were introduced in a 2nd step of the regressions: gender, performance, and the average level of demand of the degree. The adequacy of the regression model was analyzed based on its fit (significance of *F* in the linear regression and Chi-square in the logistic regression) and its explanatory power on the variance of psychological well-being (R^2^). The predictive capacity of the variables is presented using standardized values (standardized *β* for linear regression and OR for logistic).

## 3. Results

### 3.1. Description of the Sample

A total of 1287 bachelor’s, master’s, and doctoral university students participated in this study. Regarding sex, 74.6% of the sample were women and the remaining 25.4% were men. The most frequent age range was 18–21 years (58.3%), followed by 22–25 years (30.2%). As would be expected for university samples, the percentage of older participants was very low. As for the educational level, 93.2% were studying for a bachelor’s degree, 4.7% for a master’s degree, and 2.1% for a doctorate. The participants were studying for more than 15 different degrees, the most represented being psychology (18.8%), nursing (16.6%), and medicine (13.4%). In terms of academic performance, the majority of participants (around 63.4%) had average grades between 6.5 and 8.4, with scores above or below this range being infrequent. The average level of demand of the degrees studied by the participants was 11.10 (i.e., medium–high demand). The mean scores on the perfectionism (M = 27.57) and well-being (M = 89.39) scales were in line with the normative scores commonly obtained on both measures. Finally, 31.8% of the sample reported having suicidal thoughts in the last year.

As for gender differences, male students showed a slightly higher academic performance than female students (*V* = 0.13), studied more demanding degrees (*d* = 0.53), were slightly more perfectionist (*d* = 0.15), and showed worse psychological well-being in two dimensions: positive relations with others (*d* = 0.21) and personal growth (*d* = 0.23).

### 3.2. Relationship between Perfectionism, Performance, Demandingness, Well-Being and Suicidal Ideation

Table 1 shows the correlations between academic perfectionism, academic performance in terms of the average grade, the level of demand of the degree studied, and the different dimensions of psychological well-being included in the BSPWB-A. The pattern of correlations in men and women was almost identical, although the intensity of the relationships showed a slight variation. In this way, academic perfectionism correlated positively with performance and negatively with all the different dimensions of the psychological well-being scale: the total score, self-acceptance, positive relations with others, autonomy, and personal growth. On the other hand, perfectionism did not show a significant relationship with the level of demand of the degree.

In order to analyze the differences in the level of academic perfectionism in relation to having experienced suicidal thoughts in the last year, t-tests were conducted. In men, there were no significant differences (*t* = −1.84; *p* = 0.06; *d* = 0.23) in the level of perfectionism between participants who had not experienced suicidal thoughts and those who had (M of 28.38 and 30.05, respectively). In females, the mean score in academic perfectionism among those who had reported suicidal thoughts was significantly higher (*t* = −7.66; *p* = <0.001; *d* = 0.55) than that of the participants who had not (M of 29.99 vs. 26.01).

### 3.3. Predictors of Psychological Well-Being

Table 2 shows the results of the linear and logistic regression conducted to analyze the predictive capacity of academic perfectionism, performance, the level of demand of the degree, and gender (IVs) on psychological well-being and suicidal ideation in the last year (DVs). The first section includes the independent terms (or main effects) while the second section includes the interaction between perfectionism and the other three VIs (i.e., performance, demandingness, and gender). As shown, two of the IVs in the model had a significant direct effect on psychological well-being: the level of perfectionism (*β* = −0.388) and performance *(β* = 0.131). When the interaction terms are introduced (step 2), their significance disappears and two interaction terms reach significance: perfectionism interacting with the average grade (*β* = 0.364) and with the level of demand of the degree (*β* = −0.461). Figure 1 and Figure 2 show the impact of these interactions in predicting psychological well-being. As can be seen in both cases, psychological well-being decreases as the level of perfectionism increases; however, the reduction in well-being is greater among participants with a poorer performance (i.e., ineffective perfectionists) and among those taking high- or medium-demanding degrees (i.e., perfectionists in demanding environments).

As for the logistic regression (Table 2), step 1 showed that two of the variables introduced had a significant predictive capacity on suicidal ideation in the last year: higher academic perfectionism is related to the higher presence of suicidal ideation (OR = 1.075) while that higher academic performance is related to lower suicidal ideation (OR = 0.900). Unlike the previous regression, none of the interaction terms in the equation showed a predictive ability on suicidal ideation.

## 4. Discussion

This study aimed to analyze the extent to which high levels of perfectionism can compromise the psychological well-being of those who exhibit them to the point where they become associated with self-destructive ideas. A first noteworthy aspect of the findings is that one-third of the participants (31%) showed suicidal thoughts in the last year. Taliaferro et al. [22], in their research conducted with Asian and American university students, reported a lower prevalence (18.2%). These high percentages are an indicator of the mental health of university students. In Spain, the competitive environment faced by university students could explain the high levels of stress and the high prevalence of emotional and anxiety disorders identified in university students [23,24]; these disorders have been related to suicidal ideation [25]. Likewise, other studies find the cause of these suicidal thoughts in the widening gap between ambitions and the perceived lack of success experienced by students due to the inadequate organization of the academic system that makes them feel less successful as they strive harder [26].

The males who participated in the present study may have experienced or at least perceived this pressure to a greater extent since, compared to females, they scored higher in perfectionism, academic performance, and degree demandingness and displayed worse emotional well-being in terms of personal growth and positive relations with others. This means that the male sub-sample studies more demanding degrees (which have higher admission cut-off scores), have higher average grades, and perceive that they have low personal growth, and that their interpersonal relationships are less satisfactory than expected. Contrary to these findings, some studies report that females tend to achieve slightly higher grades than males. As presented by Chui and Wong [18], it may be that, in certain cultures, parental expectations for sons are still higher than those for daughters, so male students may have internalized higher levels of pressure to meet and exceed very high expectations. When such standards cannot be achieved, there is a tendency to self-isolate, which would explain the low levels of satisfaction towards building quality interpersonal relationships and the feeling of self-improvement and personal growth [27]. Likewise, more perfectionistic individuals seem most vulnerable to real and imaginary external criticism, in such a way that, on many occasions, they prefer to avoid contact with close peers and manifest themselves as more solitary individuals [26]. Despite this, this should be addressed in future research to confirm such gender effects, as other international studies have found no differences between men and women in perfectionism levels [4].

An in-depth analysis of the existing relationships between the variables evaluated shows that there is a direct and significant relationship, both in men and women, between perfectionism and academic performance, a highly expected aspect that was already reported in previous studies without finding differences based on gender [4]. Thus, students who reported high levels of perfectionism presented a significantly more outstanding academic performance than non-perfectionists, which has been explained by the greater intrinsic motivation, the high performance standards, and the more adaptive learning strategies that seem to be presented in those who are more perfectionist [9]. These individuals tend to be highly motivated, are ambitious in completing their tasks and committed to their work, and focus on internal control by thoroughly evaluating themselves to achieve perfection and avoid mistakes [8]. However, excessive levels of self-criticism can lead to emotional exhaustion and increase one’s vulnerability towards various psychological disorders. In the present study, a significant but inverse relationship was found between perfectionism and psychological well-being, findings consistent with those of other research [19], although in this case, the pattern shown by men and women is not the same. While in the case of men, psychological well-being in its personal growth aspect is not compromised; for women, all the dimensions of psychological well-being are compromised. This may be because boys perceive tasks oriented to academic performance as activities aimed towards continuous personal development. Therefore, the additional efforts they must make to reach their academic expectations and achievements are perceived as the time they also spend on self-knowledge and personal improvement, which positively impacts this area of their psychological well-being.

As for academic performance, when referred to men, the higher the grades, the greater the self-acceptance, while in the case of women, the same relationship is not observed. Following the line of the previous interpretation, men seem to assume more frequently that high levels of performance are associated with a way of living that they willingly accept, so their mental health appears to be more protected than that of girls, in whom self-acceptance is not significantly correlated with high grades. Hu et al. [28] considered that students may experience shame and rejection towards themselves due to not meeting the academic standards they set for themselves, which suggests that high achievers who reach the proposed expectations will evaluate themselves positively and accept their good and bad qualities more easily.

On the other hand, unlike what was found in the male subsample, those women who reported having had suicidal thoughts in the last year tend to report higher levels of academic perfectionism compared to those who have not had this type of suicidal ideation. The literature relating perfectionism and suicidal ideation is extensive and suggests that suicide may be the end result of a process that has its starting point in perfectionism [26]. These studies state that people with perfectionist tendencies, by being more self-critical, may be more susceptible to maladaptive behaviors and absolutist all-or-nothing thoughts, characteristic among those with depressive disorders, which lead them to interpret failures as a catastrophe and to present suicidal thoughts more frequently. In this sense, given that women are more likely than men to present the symptoms of depression and anxiety, those who present perfectionist thoughts are more likely to do so maladaptively and foster suicidal ideation [29].

Beyond these gender differences and considering the sample as a whole, the two variables that have been found to be able to explain psychological well-being are perfectionism and academic performance, and the interactions established between these two variables, as well as between perfectionism and the level of demand, are also relevant. Thus, the average grade during the previous year appears as a moderator between well-being and perfectionism. In this way, those who have scored lower grades are those who show lower psychological well-being when high levels of perfectionism are present, probably due to frustration. In this sense, the scientific literature establishes that the high expectations and unrealistic goal setting that characterizes perfectionists could contribute to their increased distress and discomfort, as it widens the gap between what they aspire to achieve and what they believe they have actually achieved [12]. Hence, as their performance is lower, the gap between their results and their expectations increases, which will increase their dissatisfaction with their performance due to their motivation to excel and the fact that their results are not as good as expected. This may lead them to experience feelings of frustration, preventing them from achieving their personal well-being [26].

Additionally, the level of demand of the degree is shown to be a moderator between well-being and perfectionism. In lower demanding degrees, higher levels of perfectionism are associated with good psychological well-being, while in higher demanding degrees, the well-being levels are lower when perfectionism increases. These results are in line with what has been reported in previous research with medical students [6], a degree considered to be highly demanding due to the pressure and responsibility that future doctors face from the admission exams where they must achieve high average scores in order to be admitted. These investigations point out that, given that medical students already have higher levels of anxiety due to the demanding selective process that they must go through in some countries [30], those who present a higher degree of perfectionism also experience higher levels of depressive symptoms and anxiety. However, in degrees where the pressure is lower, as is the case of future teachers, the relationship between perfectionism and life satisfaction turns out to be positive [8]. The same occurs in other degrees such as psychology [31] where perfectionist students show lower levels of academic burnout, which supports the moderating role that context pressure has on the relationship between perfectionism and psychological well-being.

Perfectionism and academic performance have also proven to help explain suicidal ideation in the last year in the analyzed sample. These results are related to those of other authors [22,26]. These studies report that striving for success and trying to perform to perfection cannot lead to suicide on their own; however, when the drive for success is combined with constant comparison with unachievable goals and a fear of failure, the strict perfectionist all-or-nothing mindset (total success being the only alternative) may lead them to experience low self-esteem and feelings of worthlessness, despite high performance, which may lead them to develop suicidal thoughts [12,26]. In this regard, O’Connor [13], in his systematic review of 29 international studies, found considerable evidence that high levels of perfectionism, expressed as adverse reactions when making mistakes, a tendency to doubt the quality of one’s performance, and excessive self-criticism, increased vulnerability towards suicidal ideation.

From the findings obtained in this study, it can be primarily concluded that high levels of perfectionism have been associated slightly with suicidal ideation in the last year and this ideation is lower when academic performance is high. These findings are significant given that this research involved a large non-clinical sample of students from a wide range of degrees, which allows the results to be generalized. Furthermore, this study has not only focused on examining the different mental health characteristics of participants, but also on examining the possible causal relationships between the variables explored. However, this study is not without limitations. First, the use of self-reports involves certain biases given that they are subject to the participants’ capacity for introspection and their interpretation of the questions, which can lead to the responses being biased by their social desirability, resulting in scalar errors. Likewise, it should be noted that the sample was collected during the COVID-19 emergency situation in our country, which impacted the regular teaching process through the introduction of new methodologies and may have influenced the results obtained. Moreover, longitudinal research that addresses whether the correlations found vary according to the stage of the student’s academic career (e.g., bachelor’s, master’s, or doctoral) or the time of the academic term (e.g., before exams vs. after assessment deadlines) would be of interest in the future.

## 5. Conclusions

This cross-sectional study makes a significant contribution by showing there is a relation between high levels of perfectionism and the pressure to which students are subjected and worse mental health states, underlining the differences associated with their gender and highlighting the prevalence of self-destructive and suicidal thoughts over the last year, and the risk that this entails. Currently, the academic society is focused on preparing excellent students that will lead them to the top of the greatest international rankings, without considering their psychological well-being. This study aims to highlight the importance of not only working on the development of intervention programs that contribute to the reduction in maladaptive perfectionist thoughts and behaviors among university students but also suggests that the above results should be considered when developing academic curricula, taking into account the pressure and academic burnout to which future professionals are often subjected.

## Figures and Tables

**Figure 1 ijerph-20-00085-f001:**
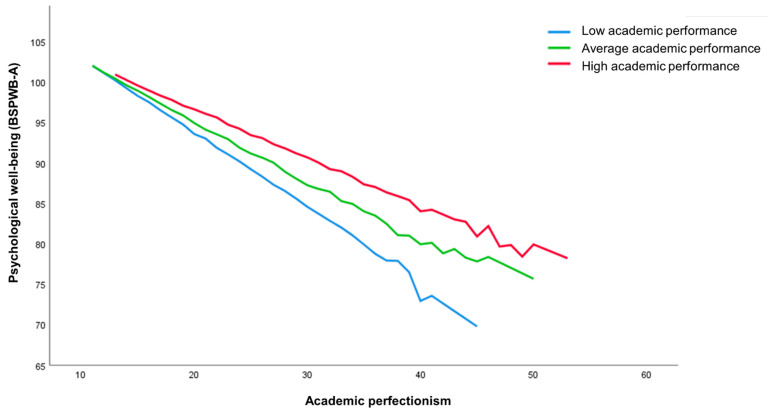
Academic performance (average grade) as a mediator between the level of perfectionism and psychological well-being.

**Figure 2 ijerph-20-00085-f002:**
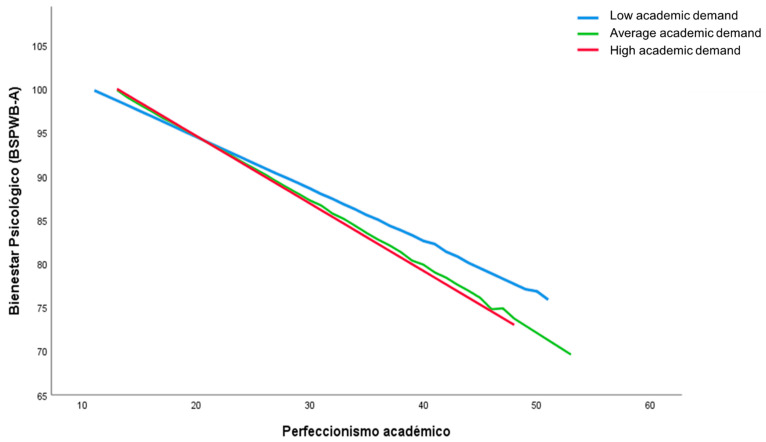
Level of academic demand of the degree as a mediator between the level of perfectionism and psychological well-being.

**Table 1 ijerph-20-00085-t001:** Correlation between academic perfectionism, performance, demand, and psychological well-being.

	1	2	3	4.1	4.2	4.3	4.4	4.5
1. Academic perfectionism (SPS ^†^)		0.259 ***	−0.024	−0.366 ***	−0.246 ***	−0.289 ***	−0.389 ***	−0.107 **
2. Academic performance (average grade)	0.292 ***		0.014	−0.010	0.047	0.036	0.038	0.014
3. Level of demand of the degree	−0.002	0.081		−0.030	−0.014	−0.014	−0.032	−0.028
4.1 Psychological well-being (BSPWB-A ^‡^, total score)	−0.301 ***	0.112	−0.038		0.843 ***	0.709 ***	0.844 ***	0.606 ***
4.2 Psychological well-being (BSPWB-A ^‡^, self-acceptance)	−0.193 **	0.155 **	−0.033	0.880 ***		0.404 ***	0.679 ***	0.445 ***
4.3 Psychological well-being (BSPWB-A ^‡^, positive relations with others)	−0.340 ***	−0.008	-0.031	0.722 ***	0.489 ***		0.374 ***	0.312 ***
4.4 Psychological well-being (BSPWB-A ^‡^, autonomy)	−0.324 ***	0.109	−0.037	0.843 ***	0.691 ***	0.410 ***		0.378 ***
4.5 Psychological well-being (BSPWB-A ^‡^, personal growth)	0.059	0.095	−0.007	0.591 ***	0.509***	0.252 ***	0.331 ***	

**Note**: The correlations for women are shown above the diagonal, while those below correspond to the sample of men; ** *p* < 0.01; *** *p* < 0.001; ^†^ SPS = Study-related Perfectionism Scale; ^‡^ BSPWB-A = Brief Scale of Psychological Well-Being for Adolescents.

**Table 2 ijerph-20-00085-t002:** Predictors of psychological well-being (linear regression) and suicidal ideation (logistic regression).

	Psychological Well-Being (BSPWB-A) ^‡^				Suicidal Ideation			
	*β*	*p*	*F*	R^2^	OR ^§^	*p*	*X* ^2^	R^2^
**Step 1: Main effects**			46.57 ***	13.7%			61.16 ***	5.5%
Academic perfectionism (SPS ^†^)	-0.388	< 0.001			1.075	< 0.001		
Academic performance (average grade)	0.131	< 0.001			0.900	0.003		
Level of demand of the degree	−0.041	0.149			0.891	0.149		
Sex	−0.011	0.702			0.754	0.702		
**Step 2: Main effects + interaction**			28.21 ***	14.3%			66.38 ***	5.9%
Academic perfectionism (SPS ^†^)	−0.183	0.402			1.022	0.763		
Academic performance (average grade)	−0.105	0.315			1.056	0.696		
Level of demand of the degree	0.187	0.092			0.835	0.281		
Sex	0.028	0.805			0.321	0.087		
Academic perfectionism X Academic performance	0.364	0.017			1.041	0.070		
Academic perfectionism X Level of demand of the degree	−0.461	0.030			0.944	0.226		
Academic perfectionism X Sex	−0.046	0.699			1.005	0.362		

**Note:** *** *p* < 0.001; ^†^ SPS = Study-related Perfectionism Scale; ^‡^ BSPWB-A = Brief Scale of Psychological Well-Being for Adolescents; ^§^ Higher than 1 favoring suicidal ideation.

## Data Availability

The data presented in this study are available on request from the corresponding author.

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
