# Peer review of "Academic Perfectionism, Psychological Well-Being, and Suicidal Ideation in College Students"

_ijerph, 2022, doi:10.3390/ijerph20010085_

Round 1

Reviewer 1 Report

Overall, an interesting study, but it does have a number of issues in interpretation and argumentation.

Moreover, contrary to popular belief, several studies have shown that a high percentage of people who commit suicide do not have a history of mental illness” – I would be highly nuanced in this. The article you refer to, is not very convincing on this part. It is so difficult really assess that. Secondly, it’s a qualitative study on the people who knew 6 men who died from suicide. I think that really isn’t enough to state your case.

The mean time spent completing 176 the survey was 9,06 minutes (SD=8.36).” Is that with the excluded also in it? Because, it might be then that people completed the survey at random, if you have a SD that large. Have you checked whether there are indications of respondents having completed the survey so fast it is unrealistic? It is quite easy to check: just look at the time they took to complete. If it’s like 1 minute or less, you know it’s impossible.

Titel of 2.3: “2.3. Stadistical analysis” – small writing error.

The suicidal ideation Was measured through” – writing error.

I wonder if it would not have been best just to look at bachelor students, considering your sample. 93.2% is bachelor students. Would it not be best to be able to make really clear conclusions on that sample instead of the constant confusion on on the small master and phd levels? Or – related – is doing a master after a bachelor not so obvious to do in Spain? In short: it might be best to give an indication of how representative this sample is. I assume that for PhD’s you might actually have a higher percentage than that there phd students in Spain? Again, I don’t know – it would be handy to know.

The suicidal ideation Was measured through a single question ("Have you ever been so 219 distressed that you have thought of death as liberation?") of dichotomous response (Yes/ No).” I wonder why the authors chose to frame it like this? Because, you are then connecting CURRENT wellbeing with life-time suicidal ideation. I think that can lead to wrong assessments, because you want to know the current relationship, no?

Secondly, I’m no suicide expert, but if I’m not mistaken, the reasons for contemplating suicide are quite diverse, and not just liberation. So, why was it framed like this?

A first noteworthy aspect of the findings is 328 that one third of the participants (31%) showed suicidal thoughts. This very high percent- 329 age offers a first indicator of the mental health of university students subjected to high 330 stress levels, which probably originated intrinsically.” This is misleading, considering how it was assessed. You don’t know if they still have such thoughts.

As for the logistic regression (Table 2), step 1 showed that two of the variables 301 introduced had significant predictive capacity on suicidal ideation: academic 302 perfectionism (OR=1.075) and performance (OR=0.900)
Very small though, no? It is almost meaningless. You can also see that in your pseudo Rsq value here.

Also, this means that the higher the performance, the smaller the chance on suicidal ideation. So, I don’t understand your conclusions… The higher their grade, the lower the chance of having ever had suicidal ideation. That is what I read into this odds ratio. So, your conclusions are I think wrong.

From the findings obtained in this study, it can be primarily concluded that high lev- 445 els of performance associated with perfectionism clearly affect mental health as measured 446 by two parameters, namely, perceived lack of psychological well-being and suicidal ide- 447 ation.

This study makes a significant contribution by showing the effect that the high levels 465 of perfectionism and the pressure to which students are subjected affect their mental 466 health, underlining the differences associated with their gender and highlighting the prev- 467 alence of self-destructive and suicidal thoughts and the risk that this entails.

You did not show an effect, you showed an association. This is a cross-sectional study. Please remove any instance of “effect”, “impact”, et cetera. I might be that their mental health affects their grades.

You should also clearly indicate that this is a cross-sectional study.

What pseudo Rsq did you use for the logistical regressions?

Reviewer 2 Report

This is an interesting study looking at how mental health issues in Spain have
become one of the biggest issues affecting the health of younger generations,
especially when it comes to suicide. In addition, the impact of certain socio-demographic
variables, such as gender differences in this dimension, is discussed. Over the past decade, a number of papers have been published on
the relationship between perfectionism and suicidal ideation, and the authors have made some good findings
However, the reviewer has the following suggestions and questions to ask:

1 Please explain why the sampled cases focus on some certain departments such as medicine, nursing, and psychology, and whether there is any bias in the results.

2 The female sample were three times larger than the male sample, why?

3 The samples come from different majors, how did the authors compare their academic performance? Since different professionals have their own "definition of good ranking"?

4. In the discussion session, the author showed more gender differences, and lacked in-depth discussion on the rest of the findings. For example, one-third of the participants (31%) showed suicidal thoughts. This very high percentage, could the authors add more discussion about this finding?

Round 2

Reviewer 1 Report

I thank the authors for their answers And adjustment.